# Small Gastric Stromal Tumors: An Underestimated Risk

**DOI:** 10.3390/cancers14236008

**Published:** 2022-12-06

**Authors:** Jintao Guo, Qichao Ge, Fan Yang, Sheng Wang, Nan Ge, Xiang Liu, Jing Shi, Pietro Fusaroli, Yang Liu, Siyu Sun

**Affiliations:** 1Department of Gastroenterology, Shengjing Hospital of China Medical University, Shenyang 110004, China; 2Innovative Research Center for Integrated Cancer Omics, Shengjing Hospital of China Medical University, Shenyang 110004, China; 3Gastroenterology Unit, Hospital of Imola, University of Bologna, 40126 Imola, Italy

**Keywords:** gastrointestinal stromal tumor, next-generation sequencing, small GIST, endoscopic resection

## Abstract

**Simple Summary:**

In this study, the high oncogenic mutation frequency (96%) of small GISTs is identified by whole-exome sequencing and targeted sanger sequencing in the entire cohort (*n* = 76) of a Chinese population. The BRAF-V600E hotspot mutation was present in ~15% small GISTs. Positive surgical or endoscopic resection should be considered for small GISTs because of their universal oncogenic mutation and undefined prognosis.

**Abstract:**

Background and Objectives: Small gastrointestinal stromal tumors (GISTs) are defined as tumors less than 2 cm in diameter, which are often found incidentally during gastroscopy. There is controversy regarding the management of small GISTs, and a certain percentage of small GISTs become malignant during follow-up. Previous studies which used Sanger targeted sequencing have shown that the mutation rate of small GISTs is significantly lower than that of large tumors. The aim of this study was to investigate the overall mutational profile of small GISTs, including those of wild-type tumors, using whole-exome sequencing (WES) and Sanger sequencing. Methods: Thirty-six paired small GIST specimens, which were resected by endoscopy, were analyzed by WES. Somatic mutations identified by WES were confirmed by Sanger sequencing. Sanger sequencing was performed in an additional 38 small gastric stromal tumor samples for examining hotspot mutations in KIT, PDGFRA, and BRAF. Results: Somatic C-KIT/PDGFRA mutations accounted for 81% of the mutations, including three novel mutation sites in *C-KIT* at exon 11, across the entire small gastric stromal tumor cohort (*n* = 74). In addition, 15% of small GISTs harbored previously undescribed BRAF-V600E hotspot mutations. No significant correlation was observed among the genotype, pathological features, and clinical classification. Conclusions: Our data revealed a high overall mutation rate (~96%) in small GISTs, indicating that genetic alterations are common events in early GIST generation. We also identified a high frequency of oncogenic BRAF-V600E mutations (15%) in small GISTs, which has not been previously reported.

## 1. Introduction

Gastrointestinal stromal tumors (GISTs) are the most common mesenchymal tumors of the gastrointestinal tract and have phenotypic similarities with interstitial cells of Cajal (ICCs) [1,2]. GISTs are commonly present in the stomach (60%) and small intestine (25%) [1]. GISTs with a diameter <2 cm are defined as small GISTs, which can be subdivided into mini- (1–2 cm) and micro-GISTs (<1 cm). The annual age-adjusted incidence averaged 6.8 per 1,000,000, and GISTs are more common in males, non-Hispanics, Blacks, and Asians/Pacific Islanders [3]. Most GISTs are usually asymptomatic and incidentally discovered during endoscopy or surgery. The diagnosis and classification of small GISTs are currently based on pathological features and imaging methods, such as computed tomography and endoscopic ultrasound (EUS). The management (endoscopic resection or follow-up) of small GISTs is controversial, and there are no consensus-based guidelines [1,4,5,6]. Current clinical guidelines recommend surgical or endoscopic resection for small GISTs with high-risk EUS presentations. For other small GISTs, EUS surveillance every 6–12 months is recommended [6,7]. The current prognostic factors and risk indices for GISTs are commonly based on the modified NIH (M-NIH) classification [8], which focuses on the tumor size (2, 5, or 10 cm), mitotic index, primary tumor sites, and tumor rupture. Low-risk or benign tumors are defined as those <2 cm with a mitotic index of <5 mitoses per 50 high-power fields [9]. Most small GISTs are generally considered low risk, but the potential malignancy of small GISTs should not be ignored. A population-based epidemiological and mortality investigation illustrated that the 5-year mortality for small GISTs is 12%, and that some of these tumors might progress and become life-threatening [10].

Large cohort studies have shown that small GISTs have a high incidence in the stomach, with some of these tumors not being benign, as they are associated with worse gastrointestinal symptoms during regular surveillance [11,12,13]. Owing to the continuously increasing rate of small GIST detection and the earlier time of onset, their surveillance and management have been deemed controversial, with a lack of evidence-based approaches [14]. Moreover, an explanation of the epidemiology, risk factors, and etiology of these small tumors is lacking [15]. Previous studies have shown that the overall frequency of KIT/PDGFRA mutations (<76%) is significantly lower in small GISTs than that in large GISTs (85–95%) [13,16,17]. However, the Sanger sequencing used in previous studies was typically based on limited primers, likely leading to an underestimation of the mutation frequency of driver genes. Therefore, more advanced sequencing methods are required to profile the mutation status of small GISTs and understand their molecular basis. In this study, we aimed to investigate potential driver genes in small GISTs using whole-exome sequencing (WES) and targeted Sanger sequencing, which will contribute to an increase in the understanding of small GISTs.

## 2. Patients and Methods

### 2.1. Clinical Samples

We primary collected 40 paired small GIST samples from the gastric muscularis propria layer obtained from the lesion sample library (January 2022–June 2022) of the Shengjing Hospital of China Medical University (Shenyang, China). Paired blood samples were used as negative controls to differentiate somatic mutations using WES. The selection criteria included the tumor size (<2 cm), definite pathological diagnosis (according to the Chinese Society of Clinical Oncology (CSCO) criteria), and endoscopic resection methods (endoscopic submucosal dissection (ESD) or endoscopic full-thickness resection (EFTR)). Four patients with an actual tumor volume >2 cm or those who did not meet the pathological diagnostic criteria were excluded. For validating the sequencing results of WES, we collected another 60 formalin-fixed, paraffin-embedded (FFPE) small GIST tissue samples (June 2021–December 2021) from the Department of Pathology of Shengjing Hospital for targeted Sanger sequencing (Figure 1).

### 2.2. Ethics Statement

The study and tumor tissues for sequencing and clinical information collected were reviewed and approved by the Ethics Review Committee of Shengjing Hospital of China Medical University (No: 2022PS049K).

### 2.3. Whole-Exome Sequencing

Genomic DNA was extracted and sequenced according to standard protocols for next-generation sequencing (Novogene Co., Ltd., Beijing, China). Briefly, paired-end DNA was obtained according to the manufacturer’s instructions (Agilent Technologies). The adapter-modified gDNA fragments were enriched by polymerase chain reaction (PCR). Whole-exome capture was conducted using the Agilent SureSelect Human All Exon V5 Kit. A total of 60 MB of DNA sequences from 33,4378 exons of 20,965 samples was captured. After DNA quality evaluation, the samples were sequenced on an Illumina HiSeq PE150 for paired-end 150 bp reads. The average sequencing depth was 224×. The coverage of the target region was 99.6%, and 96.5% of the target bases were covered to a depth of at least 20×.

### 2.4. Validation of Variants by Sanger Sequencing

Sanger sequencing was used to verify the suspected somatic variants identified by WES and further determine the mutation rate in the supplementary FFPE samples. Briefly, primers were designed using Primer Premier 5, and gDNA was extracted from FFPE tissues and blood samples (Takara, 9782). The Invitrogen^TM^ Platinum^TM^ Green Hot Start PCR 2X Master Mix (Invitrogen, Carlsbad, CA, USA) was used for PCR amplification, and the PCR products were sent for automatic DNA sequencing (Takara). PCR thermocycling conditions were as follows: activation at 94 °C for 2 min, followed by 35 cycles of denaturation at 94 °C for 30 s, annealing at 55 °C for 30 s, elongation at 72 °C for 1 min, and final elongation at 72 °C for 5 min. The nucleotide sequences of primers used for Sanger sequencing are shown in Table 1.

### 2.5. In Silico Analysis

Somatic mutations were evaluated for their predicted pathogenic effects using in silico tools, including SIFT (http://sift.jcvi.org/, accessed on 1 January 2022) and PolyPhen (http://genetics.bwh.harvard.edu/pph/, accessed on 1 January 2022).

## 3. Results

### 3.1. Clinical Features

A total of 36 paired small GISTs samples were collected for WES (labeled P1–P36), and 38 FFPE samples were successfully extracted as qualified gDNA for targeted Sanger sequencing (labeled P36–P74) (Figure 1). The clinical features and mutation information for the 74 patients are presented in Appendix A. The age of the patients ranged from 30 to 75 years, with a median age of 56 years (Table 2). Primary tumor distributions showed that the fundus of the stomach (51.3%) and gastric body (39.2%) were the most frequent sites of small GISTs. Micro- and mini-GISTs accounted for 37.8% and 62.2% of the samples, respectively, most of which were classified as very low or low risk based on the modified NIH criteria. These small GISTs were diagnosed by endoscopy, EUS, and pathological presentations involving hematoxylin and eosin staining and positive immunohistochemical features, such as CD117(+) and CD34(+). All enrolled patients were predominantly treated with ESD and partly with EFTR.

### 3.2. Molecular Analysis

#### Small GISTs with KIT/PDGFRA Mutations

WES was performed to explore the genetic variation in small GISTs. Among the 36 patients, 30 (83%) KIT mutations and 1 (3%) PDGFRA mutation were identified (Figure 2A). The most common mutation area of *KIT* was exon 11 (72%), which encodes the intracellular juxtamembrane domain, whereas other mutated sites of *KIT* were related to exon 9 (8%) and exon 17 (3%). *PDGFRA* mutations accounted for only 3% (1/36) of the mutations, occurring in exon 18. Other probable driver genes selected through a comparison with a public database (Cancer Gene Census513) are shown in Figure 2A. Among the 36 small GIST samples, the most common form of missense substitutions was C > T/G > A. The distribution of KIT mutations is shown in the molecular structure diagram (Figure 2B). The only somatic mutation in the *PDGFRA* gene was a single nucleotide change in exon 18, c.2523A > T p.D842V, which is mainly involved in the activation loop. Direct Sanger sequencing was subsequently performed on 38 additional small GIST samples. Among these samples, 68.4% (26/38) contained missense mutations in KIT and 7.9% (3/38) contained missense mutations in PDGFRA. Most *KIT* mutations were detected in exons 11 and 9, similar to the results of the WES. Two cases harbored single-nucleotide changes in *PDGFRA* at exon 18, including c.2543A > C p.N848T and c.C2544A p.N848K. Another case harboring a *PDGFRA* mutation was determined to be c.1698_1712del p. S566E571delinsR at exon 12. No mutations were detected at exon 14 of the *PDGFRA* gene. In this study, the total KIT/PDGFRA mutation rate was 81% (Figure 3).

The novel KIT mutations identified in the small GISTs included c.1716_1717insCCAACA p.(Asp572delins3), c.1502_1503insTGCCTA p.(Ser501delins3), and c.1669_1670insTTC p.(W557delinsFR), all of which were verified by Sanger sequencing (Figure 3). One patient harboring double KIT somatic single-nucleotide variants, including C1652G p. (Pro551Arg) and c.T1679C p. (Val560Ala), was identified (Figure 4a). Furthermore, these two mutations have been considered tumor-promoting factors (COSM7342419 and COSM36302), whereas P551R, usually associated with colon carcinoma, has never been reported to be associated with GIST development.

### 3.3. KIT/PDGFRA Wild-Type (WT) Small GISTs

KIT/PDGFRA WT GISTs had no mutations in the hotspot regions of *KRAS* (codons 12/13/59/61/117/146), *PIK3CA* (codons 542/545/1047), *NRAS* (codons 12/13/59/61/117/146), or *AKT1* (codon 17). Nevertheless, we discovered four cases harboring BRAF mutations (V600E) among these KIT/PDGFRA WT GISTs using WES. Surprisingly, 18.4% (7/38) of the samples in the expanded FFPE samples were found to harbor the BRAF mutation (V600E), which was verified by Sanger sequencing (Figure 4b). Seven patients with the V600E mutation had micro-GISTs, and the remaining patients had mini-GISTs. These tumors had a spindle morphology, and immunohistochemistry was positive for CD117, CD34, and DOG1 but weakly positive or negative for Ki67.

### 3.4. Suspicious Oncogenic Mutations in Small GISTs

We screened samples for other known oncogenic driver genes in KIT/PDGFRA/BRAF WT tumors to identify whether other potential elements influenced tumorigenesis (P14). We filtered two oncogenes with somatic mutations that had been reported to be related to malignant tumors (Table 3). A common mutation site in SIRT6 (c. A956C) and a suspected mutation in GDF5 (c.A630T) were detected by WES.

## 4. Discussion

In this study, 74 small GISTs were collected for mutational analysis using WES and targeted Sanger sequencing. The mean age of the patients was 56 years, lower than the predominant median age at diagnosis of 65 years [18]. Risk assessment was conducted according to the modified NIH classification criteria standard, by considering the tumor size, primary sites, mitotic index, and tumor rupture. One case in this study was evaluated as high-risk, with a size of 1.5 cm, and three cases were classified as intermediate risk with sizes of 1.2 × 1.1 cm, 1.8 × 1.3 cm, and 2 × 1.6 cm. All other cases were assessed as very low or low risk. Therefore, even if the lesion was less than 2 cm, there was still the possibility of a medium-to-high risk. However, the cutoff size of small GISTs for endoscopic resection remains controversial. Fang et al. investigated the clinical course of small GISTs and demonstrated that a cutoff value of 1.4 cm is appropriate for treatment [19], and Wang et al. proposed that a tumor diameter of 1.45 cm should be the optimal cutoff value for resection, which were consistent with our other retrospective study [20] which identified that a smaller tumor diameter cutoff (1.48 cm) might have better efficacy in differentiating risk grades. Furthermore, a single-institution retrospective study of 69 patients with EUS-suspected GISTs showed that GISTs > 9.5 mm in diameter are associated with significant progression and that 23% of these patients show significant changes in size after more than 3 years of onset [21]. Currently, intensive monitoring with EUS is recommended for most small GISTs, while this is considered an economic and psychological burden for patients [22]. For the radical treatment of small GISTs, ESD and EFTR are relatively safe and effective treatment modalities that can significantly improve patient prognosis [12,22]. With the development of endoscopy, it is also feasible to conduct the genotype diagnosis of tumor cells via EUS-based biopsy in the early stage of GISTs [23,24]. Therefore, whether the current criteria for risk classification can be used to comprehensively evaluate or predict the prognosis of small GISTs needs to be further explored, and more scientific management of small stromal tumors needs further revision.

C-KIT and PDGFRA play vital roles in the occurrence and progression of GISTs [25]. Our study strongly indicated that the oncogenic mutation frequency in small GISTs might be underestimated, since the total mutation rate (96% vs. 74%) was much higher than expected, which suggested that oncogenic mutations are early molecular events in patients with GISTs. Among the 74 small GIST samples in this study, sequencing results revealed that the C-KIT mutation was predominant (76%, 56/74), and exon 11 of *KIT* was found to be a hotspot that accounted for 65% of the mutations (48/74). Moreover, the mutations occurring at exons 9 and 17 comprised 6.8% and 1.4% of all mutations, respectively. *PDGFRA* mutations occurring at exons 18 and 12 accounted for 5% of all mutations. The mutation comprising a substitution at position 842 in the A-loop of an aspartic acid (D) with a valine (V) in exon 18 confers primary resistance to imatinib and sunitinib but sensitivity to avapritinib [26,27]. Somatic mutations in *C-KIT* are usually found in exon 11, which might confer sensitivity to imatinib [1,2,28]. The second most common mutational hotspot in *KIT* is exon 9, which might confer resistance to imatinib, the first-line targeted therapy for GISTs. The molecular mechanisms underlying oncogenic mutations, such as KIT mutations concerning the Ras-ERK and PI3-kinase pathways, are therapeutic targets of GISTs [29,30].

Notably, we found that 11 of 74 cases (15%) (Figure 3) harbored malignant BRAF-V600E mutations, which had not been detected in previous studies of small GISTs. These results contradict the previous studies which reported the mutation rate of BRAF ranges from 1~4% for large GISTs [31], suggesting that BRAF-mutated tumors might represent a low-risk subtype of small gastric GISTs. A previous study reported that 54.8% of BRAF-mutated GISTs, which were classified as intermediate or high risk [32], were located in the small bowel or colorectum, whereas stomach-derived tumors tended to have a low risk. BRAF mutants generally activate MEK/MAPK and regulate the downstream factor ETV1, thereby promoting ICC proliferation and transformation into a tumor. Activating BRAF mutations are also frequently detected in some malignant carcinomas and tumors such as melanomas, promoting proliferation and drug resistance through the constitutive activation of the MAPK pathway [33]. Ran et al. demonstrated that the BRAF-V600E mutation could promote ICC hyperplasia in adult mouse models. However, this was insufficient to drive the malignant transformation of GIST unless it was coupled with other dysfunctions in tumor-associated genes, such as TP53 loss [34]. Another study showed that BRAF mutations along with TP53 disruption could drive smooth-muscle-cell-derived GISTs, rather than those derived from ICCs [35]. In addition to TP53, the loss of TP16, another tumor-suppressive gene, promotes the development of and leads to poor outcomes for GISTs with BRAF mutations [36]. Therefore, considering the high prevalence of gastric and small intestinal GISTs (60% and 20%, respectively), “secondary hits”, such as epigenetic regulation, likely participate in the progression of small GISTs to malignant tumors. For WT GISTs, we analyzed probable oncogenic mutations, namely in SIRT6 and GDF5, which are thought to play roles in the development of ICC hyperplasia or small GISTs. SIRT6 has been identified in patients with colon adenocarcinoma and is related to the promotion of DNA repair in cells with DNA damage [37]. Mutations in GDF5 are usually associated with skeletal developmental deficiency [38], which was also predicted to be disease-causing via the SIFT algorithm and Polyphen2_HVAR.

## 5. Conclusions

In this study, we demonstrated that genetic alterations are prevalent in small gastric GISTs, suggesting an underestimated risk of these small GISTs. Despite the high frequency of the BRAF-V600E mutation, these small gastric stromal tumors might be benign and represent a low-risk subtype of GISTs. Molecular analysis will be helpful to facilitate personalized medicine and settle disputes related to treatment for small GISTs.

## Figures and Tables

**Figure 1 cancers-14-06008-f001:**
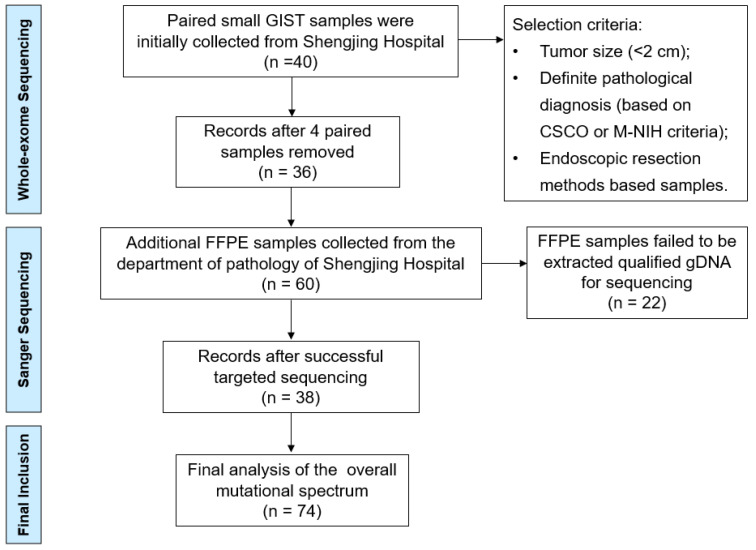
Flow chart of the selection procedure.

**Figure 2 cancers-14-06008-f002:**
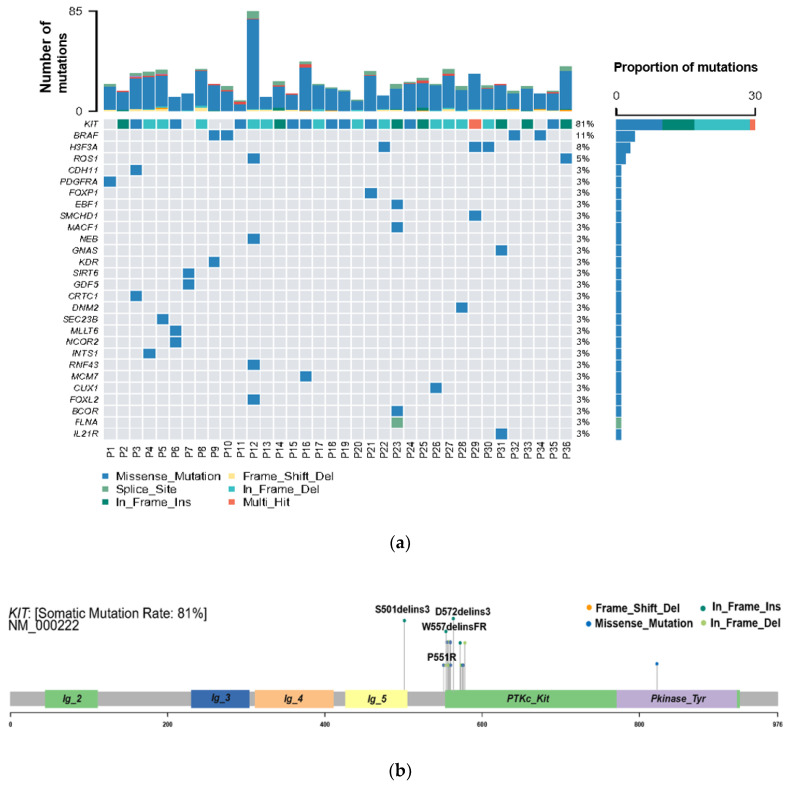
High frequency of oncogenic mutations in 36 small gastrointestinal stromal tumor (GIST) samples identified by whole-exome sequencing (WES). (**a**) Selected driver genes, by comparing somatic mutations and known driver genes in the database; (**b**) mutation distribution in the KIT molecular structure diagram, with novel mutations marked.

**Figure 3 cancers-14-06008-f003:**
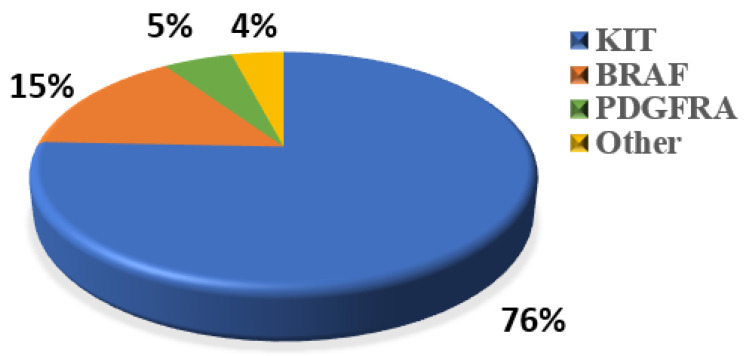
Percentage of classic oncogenic mutations in all 74 samples.

**Figure 4 cancers-14-06008-f004:**
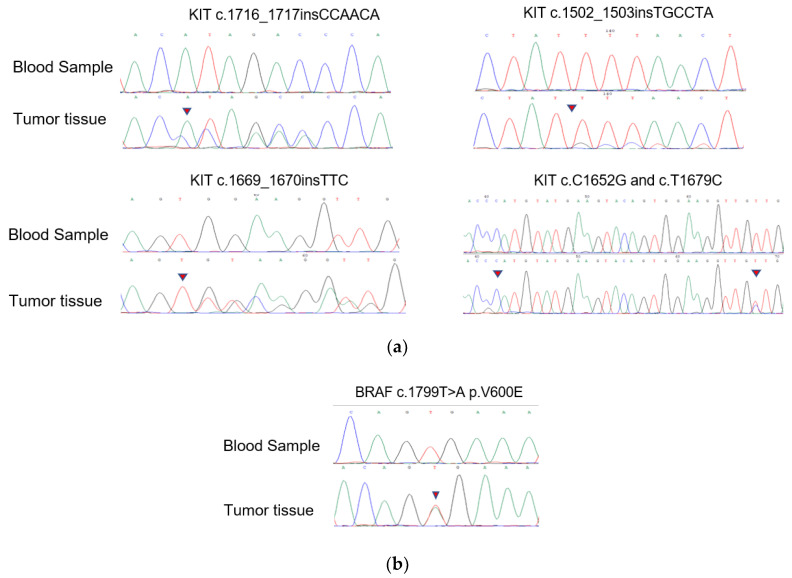
Mutation alleles based on Sanger sequencing. (**a**) Validation of KIT novel mutations using PCR-based Sanger sequencing; (**b**) validation of BRAF mutations using PCR-based Sanger sequencing.

**Table 1 cancers-14-06008-t001:** Nucleotide sequences of primers used for Sanger sequencing.

Primers	Sequence
KIT-Exon9-F	CCTTTAGATGCTCTGCTTC
KIT-Exon9-R	GGTAGACAGAGCCTAAACATC
KIT-Exon11-F	GTGCTCTAATGACTGAGACAAT
KIT-Exon11-R	AGGAAGCCACTGGAGTTC
KIT-Exon13-F	TGCATGCGCTTGACATCAGTTTG
KIT-Exon13-R	AGGCAGCTTGGACACGGCTT
KIT-Exon14-F	GTCTGATCCACTGAAGCTG
KIT-Exon14-R	ACCCCATGAACTGCCTGTC
KIT-Exon17-F	TGGTTTTCTTTTCTCCTCCAACC
KIT-Exon17-R	GCAGGACTGTCAAGCAGAG
PDGFRA-Exon12-F	TCCAGTCACTGTGCTGCTTC
PDGFRA-Exon 12-R	GCAAGGGAAAAGGGAGTCTT
PDGFRA-Exon14-F	GGTAGCTCAGCTGGACTGAT
PDGFRA-Exon14-R	GGATGGAGAGTGGAGGATTT
PDGFRA-Exon18-F	TCAGCTACAGATGGCTTGATC
PDGFRA-Exon18-R	TGAAGGAGGATGAGCCTGACC
BRAF Exon15-F	CTTCATAATGCTTGCTCTG
BRAF-Exon15-R	GTAACTCAGCAGCATCTCAG

**Table 2 cancers-14-06008-t002:** Clinicopathological characteristics of 74 patients with small gastrointestinal stromal tumors.

Clinical Pathological Characteristics	Number (%)
Sex	
Male	27 (36.5)
Female	47 (63.5)
Age	
Median, years	56
Range, years	30–75
30–50 years	22 (29.7)
51–60 years	24 (32.4)
61–75 years	28 (37.9)
Primary site	
Fundus	38 (51.3)
Junction of the fundus and body	5 (6.8)
Body	29 (39.2)
Antrum	2 (2.7)
Tumor size	
<1 cm (micro-GIST)	28 (37.8)
1–2 cm (mini-GIST)	46 (62.2)
Classification of risk	
Very low	58 (78.4)
Low	12 (16.2)
Intermediate	3 (4.1)
High	1 (1.3)

**Table 3 cancers-14-06008-t003:** Probable driver mutations of rare genes in wild-type GISTs.

Gene	Size (cm)	Nucleotide Change (c.Notation)	Amino Acid Change (p.Notation)	SIFT	Polyphen2_HVAR	Malignancy Potential
SIRT6	2 ×1.5	c.A956C	p.K319T	0.007,D	0.987,D	Low
GDF5	2 ×1.5	c.A630T	p.Q210H	0.248,T	0.395,B	Low

SIFT: sorting intolerant from tolerant; Polyphen2_HVAR: polymorphism phenotyping v2 based on HumanVar database.

## Data Availability

Data of this study can be available from the corresponding author (sunsy@sj-hospital.org) upon reasonable request approved by the Ethics Committee.

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
