# Peer review of "Small Gastric Stromal Tumors: An Underestimated Risk"

_cancers, 2022, doi:10.3390/cancers14236008_

Round 1

Reviewer 1 Report

The authors embarked on an evaluation of endoscopically resected specimens using whole exome sequencing. Somatic mutations identified were confirmed using Sanger sequencing. They came away with the conclusion that small GIST tumors have a high overall mutation rate of ~96%. This is the most significant finding in their manuscript. The authors accomplished this using WES which identified additional mutations possibly not identified using Sanger sequencing. The limitation of this manuscript is that although it nicely outlines the mutations within their cohort of patients, it is difficult to assess its clinical impact.

1. the authors noted that 15% of their samples harbored BRAF-V600E mutations.  What is the clinical significance in their cohort of patients?  It is traditionally associated with more aggressive small bowel GIST tumors. 

2.The traditional size cutoff of 1.4 cm in some institutions is appropriate for treatment.  What is the practice pattern for the authors?

3. the authors identified exon 11 as the predominant mutation in their cohort of patients. Traditionally, in large tumors, the deletion of exon 11, not necessarily missense or duplication, is associated with significantly higher potential for malignancy. Although the authors are able to verify that there are mutations in the cKit in their cohort of patients, can they expand on the comparison to traditionally larger tumors and verify the mutation is a pathogenically significant mutation?

Reviewer 2 Report

I appreciate the manuscript, “Small Gastric Stromal Tumors: An Underestimated Risk” the authors hightligth the use of the surgical or endoscopic resection should for small GISTs in the light of their universal oncogenic mutation that can be found with the use of whole exome sequencing (WES).

Minor Concerns:

 1)         In line 179in table 1 the authors  explain that” Clinicopathological characteristics and mutation information of 74 patients with small GISTs. The risk classification was according to the modified NIH criteria.

Can the authors cite an articles that describe this methods? Personally I think that the table was prepared with Sanger Sequencing in the light of that they not found mutation in some patients example P74.

2)         In line 300-304 “ The mutation comprising a substitution at position 842 in the A-loop of an aspartic acid (D) with a valine (V) in exon 18 confers primary resistance to imatinib and sunitinib but sensitivity to avapritinib. Somatic mutations in C-KIT are usually found in exon 11, which might confer sensitivity to imatinib. Can the authors cite the article that explain this sentences?

3)         Personally I had found a double citation in the reference and prefer to find in the reference also the doi, is a nice mode for be perfect.

The authors must give an appropriate cleaning and rewrite the reference. Example :

 Line 390 5. von Mehren, M. et al. Soft Tissue Sarcoma, Version 2.2018, NCCN Clinical Practice Guidelines in 391 Oncology. J Natl Compr Canc Netw 16, 536–563 (2018).

 Line 396 8. von Mehren, M. et al. Soft Tissue Sarcoma, Version 2.2018, NCCN Clinical Practice Guidelines in 397 Oncology. J Natl Compr Canc Netw 16, 536–563 (2018).

Reviewer 3 Report

Thank you for the authors. The subject is interesting, but it is not easily readable and this must be improved. Here are my comments to the paper:

The nucleotide sequencies starting from line 139 would be better to be in a table.

Web sites mentioned in the lines 159-160 should be listed in references.

Table 1 should be sited in supplementary material or be presented in the form which fits in one page. This very long table is not really readable.

All tumors were from ventricle. However, this is not the real world situation. Do the authors have information of incidence of small GISTs in other places in the literature and in their hospital?

Do the authors have any information what would be the results if other locations were also included?

Figure 1 has too many different things included. Please separate them in different pictures or consolidate information. Some of the information is easily written (only) in the text.

There is also some replay, for example line 197 and 210, please consolidate.

In the table 2 should the acronyms be written open, SIFT, HVAR, etc.

Do you have any follow-up data from these patients? This would tell about the clinical meaning of these findings.

Line 306: imatinib is not chemotherapy, but a targeted therapy.

Is there any previous information about the ethnical differences in GIST tumors? As this study is Chinese, can these results be global?

Round 2

Reviewer 3 Report

Questions were answered and corrections done. No more requirements.